# Is Endophytic Colonization of Host Plants a Method of Alleviating Drought Stress? Conceptualizing the Hidden World of Endophytes

**DOI:** 10.3390/ijms23169194

**Published:** 2022-08-16

**Authors:** Roopashree Byregowda, Siddegowda Rajendra Prasad, Ralf Oelmüller, Karaba N. Nataraja, M. K. Prasanna Kumar

**Affiliations:** 1Department of Seed Science and Technology, University of Agricultural Sciences, Bangalore 560065, India; 2Department of Plant Physiology, Matthias Schleiden Institute of Genetics, Bioinformatics and Molecular Botany, Friedrich-Schiller-University, 07743 Jena, Germany; 3Department of Crop Physiology, University of Agricultural Sciences, Bangalore 560065, India; 4Department of Plant Pathology, University of Agricultural Sciences, Bangalore 560065, India

**Keywords:** drought stress, endophytes, bio-priming, water relations, hormones, antioxidants, osmolytes, epigenetic effects, metabolites

## Abstract

In the wake of changing climatic conditions, plants are frequently exposed to a wide range of biotic and abiotic stresses at various stages of their development, all of which negatively affect their growth, development, and productivity. Drought is one of the most devastating abiotic stresses for most cultivated crops, particularly in arid and semiarid environments. Conventional breeding and biotechnological approaches are used to generate drought-tolerant crop plants. However, these techniques are costly and time-consuming. Plant-colonizing microbes, notably, endophytic fungi, have received increasing attention in recent years since they can boost plant growth and yield and can strengthen plant responses to abiotic stress. In this review, we describe these microorganisms and their relationship with host plants, summarize the current knowledge on how they “reprogram” the plants to promote their growth, productivity, and drought tolerance, and explain why they are promising agents in modern agriculture.

## 1. Introduction

The cumulative impact of human life on our planet, most notably the industrial revolution, has resulted in a progressive increase in greenhouse gas production, contributing to global warming. Consequently, the climate is changing dramatically, leading to a rise in the frequency and intensity of droughts and other abiotic stresses [1,2,3]. At the same time, feeding the budding world population, which will have risen to more than 9.7 billion by 2050, is augmented by restrictions such as spatial land allocation for agriculture and further consolidation by abiotic stress [4]. In India, 42% of agricultural land is under drought conditions [5,6]. It would be challenging to sustain greater crop yields under increasing drought stress conditions in the future [7], as the quantity of fresh water for agricultural usage is dwindling [8,9,10]. Therefore, it is critical to move the paradigm toward sustainable agriculture and find answers to concerns such as water shortage and its influence on food security [11].

Drought is a well-studied abiotic stress factor, with significant ecological and agronomic consequences [12]. Plants are among the first to suffer when soil water availability has reached critical levels [13]. Drought affects plant performance at practically every stage, either directly or indirectly [14,15]. It affects seed germination, which results in late emergence and poor seedling development [16,17]. Drought stress has a detrimental influence on the growth and development of the adult plant, due to impairments in physiological and biochemical processes including the loss of turgidity, the reduction of carbon assimilation and gas exchange, oxidative damage due to the accelerated production of reactive oxygen species (ROS) [18], reduced enzymatic activities and ion absorption [19]. With an increase in drought stress, the reduction in the leaves’ relative water content (RWC) affects stomatal conductance and net photosynthetic rates [20]. Plant productivity can be reduced by up to 73% under drought stress, depending on the growth stage and intensity of the stress [21]. Plants counteract drought stress by the activation of a set of drought-related genes and biochemical and metabolic pathways.

Many drought-stress-related genes have been identified and modern agro-biotechnologies have introduced them or modified their expression levels in plants to optimize drought tolerance [22]. Besides genetic engineering, drought-resistant varieties have been developed by plant breeding technologies, in conjunction with natural resource management, to improve water use efficiency and productivity [23,24]. However, because of the intricacy of abiotic stress tolerance systems, developing new tolerant cultivars is a lengthy process, and genetically modified plants are not favorably received in some parts of the world [25,26]. Therefore, the incorporation of microbes is an alternative method for sustainable agriculture. The role of symbiotic microorganisms in the phenotypic and biochemical adjustment of plants to environmental stresses has begun to attract greater attention [27,28,29]. Using beneficial microorganisms for large-scale agricultural application requires the basic knowledge of how they interfere with the plant’s strategies to cope with drought stress, whether they provide additional cues and chemicals that help the drought-exposed plants to adapt to the stress, or whether synergistic effects from the contributions of both partners allow the better adaptation of the two symbionts to the stress.

Microorganisms have always played a critical part in agriculture’s long-term sustainability [30]. There is longstanding evidence that microorganisms such as nitrogen-fixing microbes and arbuscular mycorrhiza fungi promote plant growth and performance under stress [28,31,32]. Microbes (bacteria or fungi) that live intracellularly or intercellularly inside plants without causing diseases are known as endophytes [33]. The ubiquitous effect of these endophytes on plant health and growth is poorly understood, mainly because the process is multi-factorial and is often unique for each microbe/host combination. Phenotypically, beneficial microbes improve stress tolerance, disease resistance, and nutrient availability [34,35], and influence plant performance by delivering hormones or other chemical mediators to the host or maintaining cytosolic ion homeostasis in the host under stress [36]. All these processes require a multi-layer crosstalk between the symbionts, in which the symbiotic interaction between the two partners results in the activation of strategies to cope with the external stress, which are better than the strategies that can be provided by each partner alone. It is also important to keep in mind that successful agricultural application is only possible if both partners profit from the symbiosis when they are exposed to drought stress.

The application of endophytes in agriculture appears to be a realistic choice, and successful strategies to cope with drought have been established in the past (Table 1). Whenever the seeds and seedlings are primarily confronted with hazards [37], seed endophytes are the primary inoculum by which to combat this plethora of biotic and abiotic stressors [38]. Endophytes applied to the seeds may enhance nutrient uptake, photosynthetic activity, and pigment levels, root growth, hydration status, and the antioxidant enzyme system of the drought-exposed plants, e.g., by the activation of phytohormone signaling, or the accumulation of soluble sugars or amino acids that function as osmolites. Under drought conditions, the endophytes may boost the expression of stress-responsive genes in the host, balance the redox and metabolic state, activate the signaling pathways, stimulate the biosynthesis of peptides or secondary metabolites, or lower the malondialdehyde levels in plants [39,40,41].

The endophytes are noteworthy for their ability to colonize without apparent negative effects on plant performance and provide habitat-adapted fitness advantages to genetically distant hosts, both monocots and eudicots [42]. Endophytes transfer habitat-specific stress tolerance to plants [43] through a process known as habitat-adapted symbiosis [44]. Plant adaptation to harsh habitats occurs typically via endophytes that are already tolerant to the extreme conditions themselves; it is believed that the tolerance is somehow transferred to the host plant in a mutually beneficial association [45,46]. The host plant’s fitness benefits in terms of extreme habitats are not only caused by microbe-induced changes in the host–omics profiles but also occur also through intergenomic epigenetic mechanisms [42,47]. Very little is known as yet of how endophytes influence the host epigenetic state under drought stress. They may play a role in priming and the epigenetic alterations in the host genome. Epigenetic information can be stored in the plant, and such a plant is then prepared to respond more strongly and rapidly when confronted with the same or even a different stress for a second time, a process called “priming” [48]. There is also increasing evidence that epigenetic changes can be transferred to the next generation [49]. Alternatively, symbiotic interactions can result in the induction of silent genes in the microbial genomes via epigenetic factors that allow the synthesis of novel metabolites with stress-related biological significance for both symbionts. For instance, in the endophytic fungus *Nigrospora sphaerica*, epigenetic modifiers such as DNA methyltransferase and histone deacetylase inhibitors increased the expression of biosynthetic pathways. The accumulation of new metabolites was observed as a result of the activation of cryptic biosynthetic gene clusters in the microbes [50]. It is conceivable that the microbes synthesize metabolites that participate in drought stress responses. A crosstalk between the two partners can establish a drought stress response that is more than the sum of the responses of the two partners alone.

Some endophytes promote the growth of quite diverse plant species [51], while the ability of others to stimulate plant growth is more restricted to a specific host’s genotype [52]. An endophytic bacterium with a broad host range is *Burkholderia phytofirmans* PsJN, isolated from onion roots, which also promotes the growth of *Arabidopsis thaliana*, grapes, maize, potato, switch-grass, tomato, and wheat [53,54] making it a powerful tool in agricultural applications. Endophytes with broad host ranges could be valuable for mitigating plants’ abiotic stress, such as from drought in agriculture. To this end, the isolation and exploration of novel endophytes from extreme habitats could be useful to confer symbiotic fitness to plants by imparting habitat-adapted symbiotic properties [55].

The symbiotic interaction between microbes and plants is highly flexible. It ranges from beneficial interactions with profits for both partners, or mutualistic interactions without visible or measurable profits for the partners, to pathogenic interactions. Depending on the host and the environmental conditions, the mode of interaction between the two partners can differ and change. A beneficial interaction can turn into a pathogenic interaction when the environmental conditions change and, in particular, when stress is increasing. The longer exposure of symbiotic crops to drought often leads to a loss of crop yield because the microbes become more aggressive, struggle for their survival and propagation, and grow faster on the weakened host. This has severe consequences for the host growth/defense balance because the plant has to invest more in defense mechanisms to restrict microbial propagation. An important task for agricultural application is the identification of endophyte/host interactions with results offering a stable benefit for the host under the given agricultural and environmental conditions. The symbionts perceive environmental threats quite rapidly, responding to them by activating multiple, often independent, molecular and biochemical responses. Nature tells us that organisms living in symbiotic interactions respond better to environmental changes than the partners alone and that the vast majority of the organisms live in symbiotic interactions, often with multiple partners in the ecosystems. The rapid and flexible adjustment of the symbionts to environmental changes has advantages for agricultural applications when compared to genetically modified plants, which respond to threats by activating one or a few newly introduced genes. In this review, we describe diverse endophytes with the potential for agricultural application under drought stress and discuss the cellular, physiological, and molecular responses that are only activated in a symbiotic interaction but are not activated by the partners alone.

**Table 1 ijms-23-09194-t001:** Endophyte-assisted drought tolerance in plants.

Endophytes	Host Plant	Plant Part Used for Isolation	Inoculated Plant/s	Beneficial Features Related to Drought Tolerance	References
*Sphingomonas*	*Tephrosia apollinea*	Leaves	*Glycine max*	Increased plant dry biomass, glutathione, glutamine, photosynthetic pigments, glycine, and proline	[56]
*Burkholderia phytofirmans* and *Enterobacter* sp.	*Allium cepa*	Roots	*Zea mays*	Increased leaf area, shoot and root biomass, photosynthesis, chlorophyll content, and photochemical efficiency	[57]
*Rhizophagus irregularis*	*Zea mays*	Roots	*Zea mays*	Higher root biomass enhanced stomatal conductance, reduced oxidative damage, and enhanced hydraulic conductivity	[58]
*Embellisia chlamydospore*, *Cladosporium oxysporum*, and *Paraphoma* sp.	*Hedysarum scoparium*	Roots	*Glycyrrhiza uralensis* and *Zea mays*	Increased root biomass and the root:shoot ratio	[59]
*Periconia macrospinosa*, *Neocamarosporium chichastianum*, and *Neocamarosporium goegapense*	*Seidlitzia rosmarinus*, *Zygophyllum eichwaldii*, and *Haloxylon ammodendron*	Roots	*Cucumis sativus* L. and *Solanum lycopersicum* L.	Increase in proline and chlorophyll content, antioxidant enzymatic activities and growth parameters	[60]
*Epichloë festucae*	*Lolium perenne*	Shoots and Roots	*Lolium perenne*	Increased P uptake, plant growth, and photosynthetic parameters	[61]
*Nectria haematococca*	*Chrysanthemum indicum* L.	Roots	*Solanum lycopersicum* L.	Improvement of plant growth parameters such as the height, stem girth, leaf characteristics, biomass, and proline accumulation	[62]
*Pantoea agglomerans*	*Alhagi sparsifolia Shap. (Leguminosae)*	Leaves	*Triticum aestivum*	Increased accumulation of soluble sugars, decreased accumulation of malondialdehyde, and the degradation of chlorophyll in leaves	[40]
*Trichoderma atroviride*	*Camellia sinensis*	Leaves	*Zea mays*	Elevated activity of ROS scavenging enzymes SOD, CAT, APX, and GR and lower H_2_O_2_ levels	[63]
*Paraconiothyrium strains*, *Embellisia chlamydospore*, and *Phialophora* sp.	*Gymnocarpos przewalskii*	Roots	*Ammopiptanthus mongolicus*	Increased branch number, and higher potassium and calcium content	[64]
*Ampelomyces* sp.	*Pyrrhopappus carolinianus*	Stems and Roots	*Solanum lycopersicum* L.	Enhanced growth and yield under optimal growth conditions	[65]
*Sarocladium implicatum*	*Brachiaria* grass cultivars	Shoots and Roots	*Brachiaria* grass cultivars	Maintained plant water status and increased dry matter content (DMC) and total non-structural carbohydrate (NSC) contents	[66]
*Epichloë occultans*	*Lolium multiflorum*	Shoots and Roots	*Lolium multiflorum*	Increased water use efficiency, net photosynthesis rate	[67]

### 1.1. Drought Sensing by Plants and the Link to Plant-Interacting Microbes

Plants have to respond to drought stress at the right time and place [68]. Short-term responses to prevent water loss through transpiration from guard cells, as well as long-term adaptations to gain stress resistance at the whole-plant level, necessitate a diversity of cellular and molecular regulatory mechanisms in plants that often occur in parallel [69]. Because drought is a multidimensional stress that affects quite different and often unrelated processes in the plant (cf. Figure 1), a wide range of physiological, biochemical, and molecular reactions are activated. It is critical to distinguish primary drought stress signals (and the immediate responses induced in the plant) from secondary drought stress signals, which induce more global responses due to an overall impairment of many downstream processes caused by the water shortage. Since drought causes different effects on cells, they recognize dehydration as osmotic, oxidative, and mechanical stresses [70,71] and perceive them via different sensing systems. Symbiotic microbes have the ability to activate several of these processes in parallel in the host, which allows a faster and more precise response to drought stress, both locally and in systemic tissue after signal propagation. Receptor-like kinases, which sense stress-induced cell wall damage, mechanosensitive calcium channels, which initiate a calcium-induced stress response, and phospholipase C, a membrane-bound enzyme that is integral to osmotic stress perception, are examples of primary sensors in the perception process or of early signaling components [72]. 

Drought-generated hyperosmotic stress (Figure 1) opens the hyperosmolality-gated Ca^2+^-permeable channels (OSCA), a group of osmosensors that are best characterized in *Arabidopsis* [73]. Thor et al. [74] showed that OASCA1.3 controls stomatal closure during immune signaling; the channel is also rapidly phosphorylated and activated upon the perception of pathogen-associated molecular patterns (PAMPs). The immune receptor-associated cytosolic kinase BIK1 interacts with and phosphorylates OSCA1.3 after the application of the PAMP flg22, which results in higher OSCA1.3 Ca^2+^ channel activity [74]. This links drought stress-sensing with the immune signaling induced by microbes that interact with plants. Shifts from a beneficial to a more pathogenic interaction, due to osmotic stress, might strongly influence the activated signaling pathways and, consequently, the drought-tolerance response of the host. 

Water shortage also induces mechanical stress at the cell wall/plasma membrane junction. The yeast MID1 is a stretch-activated channel, while the plant homologs are the mechano-sensitive MCA channels [75,76]. Yoshimura et al. [75] showed that MCA2 is a Ca^2+^-permeable mechanosensitive channel that is directly activated by membrane tension. MCA proteins, which are found exclusively in plants, are involved in multiple abiotic and biotic stress responses and provide an additional link between drought stress responses and symbiotic interaction with the microbes.

One immediate and rapid downstream response to drought is an increase in cytoplasmic Ca^2+^ elevation, and both OSCA and the plant hormone, abscisic acid (ABA), utilize Ca^2+^ for signaling. As is known from other stress responses downstream of the Ca^2+^ response, the stimulation of ROS production and alterations in gene expression profiles via mitogen-activated protein kinase (MAPK3)/(MAPK6) signaling and phytohormone (ABA) biosynthesis has been documented [77,78]. Ca^2+^ activates CDPK (calcium-dependent protein kinases) and various transcription factor networks (bZIPs, NACs, WRKYs, MYB AP2/AREB, HSFs, etc.). Additional components involved in early signaling are inositol-1, 4, 5-triphosphate (IP_3_) and cyclic adenosine 5′-diphosphate ribose (cADPR). Together with ROS, they function as secondary messengers to activate downstream responses. Almost all of these signaling compounds have also been reported to be targeted by endophytes/microbes during their interaction with plants: Ca^2+^ [79,80], CDPK [81], ROS (cf. [82,83]), MAPK3/6 [84], ABA [85] or IP_3_ [86]. Thus, the drought perception and signaling systems described here are potential targets of endophytes that can control, alter, promote, restrict, or prime the host response.

Recent studies have discovered various classes of peptides involved in drought signaling and systemic signal propagation [87]. In many cases, drought perception starts as a decline in the osmotic potential at the roots, before the information is systemically spread within the entire plant body. Clavata/embryo-surrounding region (CLE) peptides are encoded by huge gene families in all plant species. The *CLE25* gene responds to drought stress and is mainly expressed in the root vascular tissue. CLE25 activates *NCED3* expression in the root and ABA accumulation in the shoot. Thus, this peptide perceives drought in roots and activates long-distance signals, resulting in ABA-mediated stomata closure in the leaves [87]. Grafting experiments have demonstrated that CLE25 is transported from the roots to the leaves via the vascular tissues, and a CLE25-specific transporter has been postulated for *Arabidopsis* (cf. [70]). Small molecules are ideal candidates for systemic signal propagation within the plant body and are potential targets for beneficial microbes. A recent review by Roy and Müller [88] summarized the importance of peptides under nutrient stress and plant-microbe interactions, and also described how a microbial peptide mimics or alters the host physiology to enhance colonization. The reannotation of bacterial genomes might help to identify new peptide candidates, connecting drought-tolerance responses in the host with symbiotic microbes.

Not surprisingly, miRNAs which control many defense and stress response genes, participate in the fine-tuning of drought responses, as demonstrated in rice [89]. During the establishment of symbiosis, the majority of pathways targeted by miRNAs for plant defense are turned off that would otherwise have obstructed the proliferation of endophytes [90]. However, the drought-related miR396 and miR159 were significantly induced by the endophytic fungus *Piriformospora indica* in rice under drought, demonstrating that endophytes have specific effects on plant’s miRNA levels, controlling drought-tolerance traits [91]. 

An attractive hypothesis is that the aquaporins participate in osmoregulation, although sufficient evidence to support this is still missing [92]. Aquaporins facilitate the movement of water and small molecules through different membranes. A plasma membrane-localized aquaporin is phosphorylated by a CDPK and is regulated in a turgor-dependent manner [93]. The authors showed that drought lowers the apoplastic water potential, thereby decreasing aquaporin phosphorylation, lowering plasma membrane water permeability, and minimizing water loss [93]. Antisense plants with reduced plasma membrane aquaporin expression compensated for reduced water permeability through the plasma membrane by increasing the size of their root system [94]. While these results clearly demonstrate the importance of aquaporins in water uptake, their contribution to osmotic adaptation under stress requires additional study [92]. However, Gond et al. [95] reported that the rhizobacterium *Pantoea agglomerans* induces salt tolerance in tropical corn and that this is associated with higher expression levels of specific aquaporin genes.

An increase in osmoregulatory molecules, which stabilize cell membranes and maintain cell turgor, is another physiological response that plants make when they are stressed due to a lack of water [96]. Proline and other amino acids, soluble sugars and sugar alcohols, or quaternary ammonium compounds are osmoprotective compounds that accumulate in response to many forms of environmental stress, including drought [97,98], and restore the osmotic potential of the cytoplasm to drive water uptake and maintain cell turgor. These osmolytes also help to mitigate the effects of oxidative stress [99]. The above-mentioned transcription factor network regulates the expression of different stress-related genes, including those for the biosynthesis of proline, glycine betaine, soluble sugars, late embryogenesis abundant (LEA) proteins, aquaporins, heat shock proteins, and dehydrins, which, in turn, act as chaperones to provide drought tolerance [78,100]. For instance, *P. indica* alleviates NaCl stress in tomato plants by stimulating the accumulation of proline and glycine betaine [101].

### 1.2. Seed/Plant-Interacting Microbes as Meta-Organism

Seeds include a colony of endosymbionts that perform several important functions for developing seedlings, and seedlings are less likely to survive without these symbiotic microbes [104,105]. Mycovitality [106] refers to the symbiotic connections between seeds and endophytic fungi, and it is a major and ongoing field of study [107,108]. Seeds from a wide range of plants, including monocotyledons, dicotyledons, and herbaceous plants, have been found to contain diverse types of endophytic microbes [109,110]. The seed microbiome provides an immediate source of microbial aid for seedling establishment, which is especially significant for invasive plant species, when finding a compatible microbiome partner is difficult [111]. Seed-borne bacterial and fungal endophytes have a significant impact on seed germination under unfavorable environmental conditions [112] and are often crucial for the nutrient supply of the young growing seedlings. Seed endophytes are mostly mobilized via vascular connections or vertical transmission [111]. The plant/microbe community can be considered as a complex multi-genomic entity, in which the partners form a network of interspecific interactions [113] in the rhizosphere, endosphere, or phyllosphere, and the endophytes can invade the seed and plant tissues at any time during their life cycle. 

### 1.3. Habitat-Adapted Symbiosis of Endophytic Fungi

“Habitat-adapted symbiosis” refers to the association of plants with endophytes (fungi or bacteria) that help plants to adapt to climatic conditions in a habitat-specific manner, more especially by class II non-claviceptaceous endophytes from plants adapted to geothermal soils, coastal beaches or agricultural fields [42]. Endophytic fungi that have been isolated from plants in harsh environments can confer tolerance to non-host and sensitive plant species and are, therefore, of agricultural relevance [114,115].

The capacity of endophytes to colonize non-host plants and impart habitat-specific stress tolerance to their new hosts shows that symbiotic interaction may play an important role in the plant’s stress tolerance. Fungal endophytes can provide host plants with habitat-specific stress resistance [116]. For instance, drought and salinity stress tolerance [117] and thermo-tolerance [46] are conferred on non-host plants by fungal endophytes from desert plants. Besides the better adaptation of the host –omics pattern to drought, habitat-specific symbiosis may also activate an inter-genomic epigenetic mechanism [42,118].

Analysis of the endophytes from plants growing in different habitats [119,120,121] has revealed many characteristics of their agricultural application, including their widespread distribution, sustained presence in plants, and non-pathogenic nature [114]. Although they have a high potential for crop improvement, only a few species have been isolated, identified, and characterized to date. Furthermore, the available information has uncovered that the taxonomy of endophytes is quite diverse [122].

### 1.4. Seed Bio-Priming with Endophytes

The methods of inoculation have an impact on endophyte colonization [123] and different protocols for inoculation can be found in the literature [124]. Seed inoculation and soil inoculation are the two most commonly used approaches [125,126]. For seed inoculation, researchers co-cultivate prepared liquid inoculum with seed or seedlings in the early stages of their emergence from the seed coat. For soil inoculation, the inoculum is introduced into the root media or soil in pots. Pruned-root dip [127], spot inoculation [128], and foliar sprays [129] are typically used for this procedure. A meta-analysis indicates that seed inoculation is a widely utilized approach for endophyte inoculation under non-stress conditions [130], while the introduction of endophytes under stress conditions is mainly performed by soil application [131]. One well-established technique is seed bio-priming with mycorrhizal fungi, which improves seed germination, seedling survival, and seedling establishment time [132]. The inoculation method is determined by the convenience of the experiment and the efficiency of establishment of a successful symbiosis.

### 1.5. Plant-Endophyte Entity

The plant–microbe symbiosis creates an information-processing entity with complex processes of communication [133]. The contribution of the associated microorganisms and their investment in information conditioning are often ignored. Endophytes can be considered an indispensable and integral part of the plant system.

The outcome of this symbiotic interaction is mostly attributable to genetic control by the host, while the contribution of the fungal partners is often neglected. Vertically transferred core microbial species evolve with the plant host species [134] and might, therefore, contribute substantially to the response of the symbiosome to environmental cues. First, endophytic microbes have a wide range of impacts on the functioning of a plant’s micro-ecosystem and may influence the plant’s responses to numerous environmental changes [135,136]. Second, plants and microbes produce a wide range of identical metabolites with similar precursors. For example, one universal precursor for isoprenoids (carotenoids, quinones, hormones, and secondary metabolites that are important for plant defense and communication) is also produced by eubacteria and archaea [137]. Howitz and Sinclair [138] developed the xenohormesis hypothesis, which says that stress-induced chemicals from plants can be sensed by microbes, which are capable of producing identical secondary metabolites. Plants and microbes, according to this theory, possess homologous genes/gene clusters (which are likely horizontally transmitted) that might be cross-activated by the host or endophyte in an emergency (e.g., abiotic stress). Finally, indigenous endophytic microbial communities detect external cues, which detection results in an alteration of the community structure under stress. Obviously, this can have tremendous effects on the plant’s system [139,140], which responds quite differently to the altered microbiome in its tissues. These examples demonstrate that the plant’s responses to stress depend on the microbes and how they adapt to environmental changes.

### 1.6. Endophyte Services: What Are They Doing for the Plants?

Because plant–fungus associations were established early during evolution, the communication between the partners and in response to environmental cues has a long history and is highly coordinated. Waqas et al. [141] found that certain endophytic fungal isolates (*Chrysosporium pseudomerdarium*, *Aspergillus fumigatus*, and *Paecilomyces* sp.) improved seed germination, vigor index, and shoot and root growth by breaking down cellulose and providing carbon to the seedlings. Murphy et al. [142] showed that endophytic fungi stimulated seed germination and subsequent seedling growth in barley and oats. As outlined above, seed priming with beneficial fungi and bacteria can significantly improve seed germination and emergence, seedling establishment, crop growth, and yield parameters [143,144]. Although the outcome of the symbiotic interaction is often similar, the mechanisms leading to benefits for the host can be quite different and be specific to that plant–fungus association. 

The combination of the endophytic fungus *P. indica* with a nitrogen-fixing bacterium and a phosphorous solubilizer significantly increased finger-millet growth, yield, and quality [145]. Inoculation of *P. indica* on *Brassica campestris* sp. *chinensis* L. increased plant growth and carotenoids, as well as antioxidant, phenolic acid, and flavonoid levels in the host [146]. *Trichoderma harzianum* increased yield, along with the contents of chlorophyll, starch, nucleic acid, total protein, and phytohormones in maize plants [147]. Colonized rice plants showed a significant increase in the plant height, photosynthetic rate, chlorophyll content, and stomatal conductance, and also the tiller and panicle number [148]. *P**homopsis*
*liquidambari* also forms a symbiotic relationship with rice, assisting in nitrogen accumulation and uptake, which results in the promotion of growth and yield [149]. A comparative investigation of these symbiotic interactions is required to understand whether the beneficial achievements for the plants are the result of a convergent co-evolution of the two partners. 

Rehman et al. [150] investigated the seed priming of wheat seeds with an endophytic *Pseudomonas* sp. strain that increased growth and the zinc status of the host. While seed priming increased the yield, soil and foliar application boosted protein content, along with zinc concentration in the aleurone layer, endosperm, and total grain. Inoculation with *P. indica* also increased the photosynthetic electron transfer activity, as well as the accumulation of proteins that protect the primary metabolism, photorespiration, transporters, energy regulation, and autophagy. In an experiment with *Brassica oleracea* var. *acephala* (kale), Poveda et al. [151] reported that the aerial biomass of plants infected with *Fusarium* sp. and *Acrocalymma* sp. was doubled, compared to uncolonized control plants. Morsy et al. [65] isolated fungal endophytes from plants in high-salt areas and drought-stressed habitats and showed that they confer an abiotic stress tolerance to tomato plants. In comparison to non-symbiotic tomato plants, endophyte treatment provided salinity stress resistance and increased root biomass and fruit yield in both pot and field studies. In some cases, endophytic fungal colonization reduced the host´s growth, which can be explained as a form of weak parasitism [152], or as the induction of host resistance, resulting in the allocation of carbon to the production of expensive defense compounds rather than to vegetative growth [153]. Thus, the endophyte can adjust the host´s growth/defense balance according to changes in the environmental conditions.

### 1.7. Endophyte Elicitation of Drought Tolerance

Endophytes promote the growth and development of host plants either directly or indirectly (Figure 2) by secreting growth-promoting chemicals such as phytohormones, nutrients, the production of siderophores, phosphate solubilization, and starch hydrolysis [154]. One of the best-known illustrations of endophytic fungi conferring stress tolerance to hosts is described for *P. indica*. Upon the inoculation of *Arabidopsis* and Chinese cabbage, the plants displayed drought tolerance by expressing drought-stress-related genes [155]. Endophyte-inoculated wheat plants produced seeds that germinated faster in the second generation than endophyte-free plants under heat and drought stress, which resulted in increased plant height, seed weight, and water-use efficiency in pot experiments [156]. Similarly, Bodhankar et al. [157] showed increased maize growth under drought stress with the presence of the endophytic isolates *Corynebacterium hansenii* and *Bacillus subtilis*.

Besides improvements in root and shoot fresh weight, photosynthetic rate, and stomatal conductance, the endophytes stimulated the production of osmolytes such as sugars (fructans, glucose, and sucrose) and amino acids (glutamic acid and asparagine) under drought stress. 

Endophyte-inoculated plants also have high concentrations of gamma-aminobutyric acid. Wheat seedlings with the endophytic strain, *Pantoea alhagi*, isolated from *Alhagi sparsifolia*, grow better under drought and have higher sugar and lower malondialdehyde concentrations [40]. 

*P. indica* colonization readjusts plant metabolites and proteomes to increasing stress and maintains the number of aquaporins in drought-stressed plants [161]. Sandhya et al. [162] found that some endophytic bacterial strains confer drought tolerance up to 1.02 matric potential. Drought stress was alleviated in the grass *Brachypodium distachyon*, colonized by the bacterium *Bacillus subtilis* B26, which increased the expression of the stress-responsive genes [39]. Here, we summarize a few of the main targets of the microbes that are often observed when they promote drought tolerance in plants. 

 *(a)* 
*Hormonal regulation*


The most prevalent phytohormones involved in stress adaptation are abscisic acid (ABA), ethylene, auxin, cytokinin, and gibberellin (GA). The phytohormones described in the context of plant/microbe interaction are salicylic acid (SA), jasmonic acid (JA), nitric oxide (NO), nitrogen dioxide (NO_2_), strigolactone (SL), and brassinosteroids (BR). Under drought stress, ABA and ethylene cause stomatal closure and counteract auxin and cytokinin, which promote stomatal opening [163]. Hormones such as ethylene, auxin, GA, or cytokinin can be produced by the endophytes and delivered to the hosts [164,165]. They not only drive plant cell development but can also induce nutrient release to the endophyte, which promotes its growth in the host [166]. Waqas et al. [167] found that soybeans inoculated with the phytohormone-secreting *Galactomyces geotrichum* endophyte increased macronutrient absorption. The auxin transport capability of a *Pseudomonas* strain improved *Arabidopsis* root architecture [168]. Auxin synthesized by *Pseudomonas* sp. and *Pantoea dispersa* improved the root and root hair growth of rice seedlings [165]. Hamayun et al. [169] observed that the culture filtrate of the endophyte *Aspergillus fumigatus* improved seed germination and increased the growth of soybean plants. GA_3_, GA_4_, and GA_7_, as well as the inactive GA_7_, GA_19_, and GA_24,_ secreted by *A. fumigatus*, stimulated soybean growth. The endophytic *Bacillus amyloliquefaciens* strain from rice seeds produced GAs that not only boosted the physiology of the host [170] but also stimulated the SA and repressed the ABA levels in the host. IAA and GA from *Sphingomonas* sp. LK11 increased tomato [171] and *S. mutabilis* IA1 from Saharan desert wheat-seedling growth [172]. GA-producing endophytes have a positive impact on the growth and production of many agricultural plants. For example, the GA-secreting rhizobacterium *Pseudomonas putida* H-2-3 boosted soybean growth under dry conditions [173]. Auxin-producing endomicrobiota stimulated the production of 1-aminocyclopropane-1-carboxylic acid (ACC) synthase and increased ACC concentration, which resulted in the activation of ACC deaminase [174]. The ACC deaminase-producing *Limonium sinense* enhanced plant development under salinity [175].

Besides stomata closure, ABA protects plants from abiotic stresses by regulating multiple stress-responsive genes [176]. Salt stress increased the ABA content in plants and promoted their interaction with plants [177]. Khan et al. [178] found that salt-injured soybean plants that were colonized by *Penicillium minioluteum* had lower ABA levels than untreated plants. Thus, the association with endophytic fungi can also result in reduced ABA levels [179], which could be explained by less stress—and, consequently, less ABA—in the presence of the endophyte for the plant. In many plants, JA induces the biosynthesis of defense-related proteins and protective secondary metabolites in response to abiotic stress [180], whereas SA is a signaling hormone that aids in stomatal behavior, respiration, and the generation of antioxidants [181,182]. Endophyte infection reduces stress not only by lowering ABA levels but can also change JA and SA levels [141], which stabilizes the symbiosis. This results in stronger and more stress-resistant plants. Induced systemic resistance to abiotic stressors is increased by the symbiotic interaction of fungi and plants; the elevated endogenous SA level can buffer the harmful metabolites, as shown in the ROS [183]. As a result, increasing phytohormone production through the application of endophytes is a viable strategy for achieving sustainable agriculture under stressful conditions [184]. 

 *(b)* 
*Siderophore production, Fe^3+^ availability for the host*


Under iron (Fe)-limiting conditions, endophytic microorganisms can produce organic acids and siderophores (low molecular-weight iron-chelating chemicals) [185]. Siderophores from endophytes often have a stronger affinity for Fe^3+^ ions than phytosiderophores. Sessitsch et al. [186] showed that endophyte-colonized rice roots have higher expression levels of genes for siderophore synthesis, iron reception, and storage. Apart from delivering iron to their hosts, siderophores are thought to play an important function in biocontrol by quenching the bioavailable iron and denying access to phytopathogens [187]. Recently, Kanwar et al. [188] proposed a model of how endophytes may contribute to sufficient Fe uptake and maintaining Fe homeostasis in the host plant under osmotic stress. In terms of adjustment to growth and stress conditions, Fe deficiency-regulated transcription factors, such as the Fer-like iron deficiency-induced transcription factor (FIT), act as regulatory hubs. Proteins of the osmotic/ABA signaling pathways and of Fe signaling are directly or indirectly related to FIT. FIT, as a master regulator of Fe signaling, and ABI5 as a master regulator of ABA-signaling, crosstalk with each other and connect the two pathways. Endophytes promote the availability of Fe^3+^ ions under stress, alleviating Fe^3+^ shortage for their hosts and, thus, interfere with this regulatory process (cf. [189]).

 *(c)* 
*Phosphate solubilization and uptake*


Phosphorus (P) is often limiting to plant growth. The majority of P in soil are phosphates, metal complexes with minerals, or organic material. Microorganisms that are capable of solubilizing insoluble P in the soil present a strategy for reducing the usage of synthetic P fertilizers. The search for root-colonizing P-solubilizing microorganisms that can grow under drought stress conditions is a major field of study in agricultural research. They could be used for formulation, in particular, for crops exposed to abiotic stress [190].

Fungi are more efficient in solubilizing bound nutrients including P than are other soil microorganisms [191]. Gupta et al. [192] showed that fungal isolates solubilized Ca_3_(PO_4_)_2_ and rock phosphate more efficiently than bacteria. Furthermore, Vassilev et al. [193] found that *Aspergillus* and *Penicillium* effectively solubilize rock phosphate, which is then taken up by the plants [194]. Another example of a phosphate-solubilizing microorganism that is beneficial for long-term agriculture [195] is *Curvularia geniculate*, a dark septate endophyte that was originally isolated from *Parthenium hysterophorus* L. It can solubilize multiple P sources (AlPO_4_, FePO_4_, and Ca_3_(PO_4_)_2_) and also produce IAA when inoculated into pigeon pea (*Cajanus cajan*).

Roots possess two distinct modes of P uptake from the soil, direct and indirect uptake. The direct uptake of P is facilitated by the plant’s own P transporters, while indirect uptake occurs via symbionts, wherein the host plant obtains P primarily from its microbial partner [196]. One well-investigated example is mycorrhizal fungi. In their hosts, mycorrhiza-specific plant phosphate transporters, which are required not only for symbiotic P transfer but also for the maintenance of the symbiosis, have previously been identified [197]. The P acquisition technique of roots with root-colonizing endophytes appears to be different. For instance, *P. indica* plays a crucial role in P transfer to plants, especially in P-deficient environments [198]. *P. indica* contains a high-affinity phosphate transporter involved in improving phosphate nutrition levels in the host plant under P-limiting conditions [196]. However, the way in which the P transfer from the endophyte to the root cell occurs has not yet been understood completely. In arbuscular mycorrhizal symbiosis, the uptake of phosphate from the hyphae to the root cells occurs through specific host phosphate transporters. Their genes are highly expressed in the root cells with arbuscles. The hyphae of endophytic fungi, such as *P. indica*, are mainly found in the apoplastic space of the roots; the phosphate transporter genes of the host are not upregulated by *P. indica* colonization. Since the hyphae have better access to phosphate from the soil than the larger root cells, the higher phosphate concentration in the hyphae located in the root apoplast allows a better supply of the host with phosphate compared to roots that are not colonized.

P starvation activates several transcription factors, such as MYB, WRKY, and zinc finger transcription factors [199], which respond not only to P starvation responses but also to stress-related hormones, drought, cold, heat, or pathogen infections. They bind to regulatory elements in the promoters of P starvation-responsive genes for phosphate transporters, phosphate starvation-induced proteins, or miRNAs. The way in which signals from beneficial endophytes interfere with this regulatory circuit to reduce drought stress responses remains to be determined.

 *(d)* 
*Starch hydrolysis*


The starch–sugar interconversion in the source and sink tissues plays a profound physiological role in all plants. Dong and Beckles [200] describe how changes in starch metabolism can facilitate adaptive changes in source-sink carbon allocation, for protection against environmental stresses. Water, salinity, and heat stress repress starch biosynthesis and increase sugars in numerous crop plants. The higher sugar concentrations protect the embryo and young seedlings against osmotic stress by functioning as osmolytes and most likely activate multiple processes that strengthen the cell and support it in overcoming drought stress. Microbial and, in particular, fungal enzymes that participate in starch degradation and the accumulation of soluble sugars participate in strengthening the plant under drought stress.

Amylases are starch-degrading enzymes that catalyze the hydrolysis of polysaccharide internal glycosidic linkages. The majority of the described microbe-derived amylases come from soil fungi, such as *Aspergillus, Penicillium*, and *Rhizopus* [201], although only a few publications describe amylases from endophytic fungi. The endophytic fungi *Gibberella pulicaris*, *Acremonium* sp., *Synnematous* sp., and *Nodilusporium* sp. produced raw starch-degrading enzymes [202]. Maria et al. [203] found that a few endophytic isolates from the mangrove angiosperm, *Acanthus ilicifolius* L., and the mangrove fern *Acrostichum aureum* L. produced amylases. The additional degradation of starch by fungal enzymes under osmotic stress conditions provides sugar monomers that strengthen the plant at the expense of reserve storage [200]. How the fungal enzymes gain access to the plant’s amylose stores remains to be analyzed.

 *(e)* 
*Photosynthetic capacity*


Photosynthesis is most vulnerable to water deficit and is directly related to the accumulation of biomass [204]. Endophytes promote the photosynthetic capacity of drought-exposed host plants, as shown, for example, in *Lolium perenne* [205], *Festuca* [206], *Phyllostachys edulis* [207], and *F. sinensis* [208]. The beneficial effect of endophytes on photosynthetic capacity can be associated with the reprogramming of the plant’s metabolism and the accumulation of chemical mediators that help the plants to adapt to water shortages [209].

To obtain quantitative data, chlorophyll fluorescence techniques can be used to analyze the efficiency of the electron transport through the photosystem II [210]. These non-invasive measurements allow for the rapid identification of changes in the electron flow efficiency before the pigment content in the plastids is altered. The loss of chlorophyll has more severe effects on photosynthesis and occurs later, i.e., when the water shortage has such severe consequences for the cell and plastids that damaged or intact photosystems are degraded. After longer exposure to drought, photosystem II activities are roughly correlated with the chlorophyll content (cf. [211]). In many studies, a delay in lowering photosynthetic activity or chlorophyll content was observed when the drought-exposed plants were colonized by beneficial endophytes. For instance, *F. acuminatum* significantly increased SPAD values under drought-stress conditions [212,213]. Sherameti et al. [214] demonstrated that *P. indica* confers drought tolerance to *Arabidopsis.* The decline in the photosynthetic activity of drought-exposed seedlings was strongly retarded by the fungus; this was accompanied by the stimulation of drought stress-related genes. In conclusion, photosynthetic parameters are useful tools by which to analyze whether an endophyte promotes plant performance under drought stress.

 *(f)* 
*Water relationships and relative membrane permeability (RMP)*


In many crops, leaf RWC is a key indicator of water stress and drought resistance [215]. It explains how water is used to keep cells hydrated when there is a drought. Drought has reduced RWC in both inoculated and un-inoculated plants [57]; however, the water status of the plant is improved by endophyte association, which increases RWC by up to 30% [66]. This could be due to a more effective root system in infected plants. Higher RWC also indicates less cell wall damage; the endophytic connection may aid in the preservation of cell wall structure under drought conditions. This could result in the greater ability of cells to absorb and save water than in the uncolonized controls [216]. Endophytes may protect the cell wall from dehydration-induced damage by upregulating genes involved in the production of the cell wall polysaccharides cellulose, hemicellulose and pectin, or cell wall modifiers, and cell wall proteins [217]. Besides their effects on the cell wall, drought stress accelerates the relative membrane permeability (RMP) in both inoculated and uninoculated plants; however, endophyte inoculation helped seedlings retain the RMP and reduced leaf damage by 43% [57]. A positive correlation between drought stress sensitivity and membrane damage was also observed by Vardharajula et al. [218] and Sandhya et al. [219]. 

Under drought conditions, endophyte association in *Brachiaria* cultivars (particularly cv. Tully) increases the cellular apoptosis susceptibility proteins [155], which stimulates aerenchyma formation (i.e., intercellular gas-filled lacunae) in the leaf sheath and root tissues, where it holds apoplastic water and allows for water conservation in plant cells, contributing to higher values of the leaf RWC [220,221,222]. 

 *(g)* 
*Endophytes activate the plant antioxidant system and the accumulation of osmolytes under drought stress*


Plants naturally produce ROS as a result of their normal metabolic activity [223]. ROS function as signaling molecules and participate in balancing symbiotic interactions [224]. However, plants require ROS scavengers, such as antioxidant amino acids and enzymes, to maintain homeostasis. Excess ROS cause oxidative damage to nucleotides, proteins, and lipids, which ultimately leads to cell death, resulting from a disruption of the equilibrium between ROS formation and scavenging [150]. Furthermore, each cell and subcellular compartment has its own ROS homeostasis, depending on the kind of cell and cellular compartment, the level of stress, and the ROS gene network [225].

Endophytic microbe colonization increases the plant’s levels of antioxidant enzymes such as catalase (CAT), superoxide dismutase (SOD), peroxidase (POD), and ascorbate peroxidase (APX) [116], or of non-enzymatic antioxidant compounds, such as ascorbate (AsA), glutathione (GSH), and carotenoids [226]. This results in lower ROS levels in the cell or organelle [115], since the antioxidant enzymes counteract the accumulation of excess free radicals. Under saline conditions, the levels of CAT, APX, and GR in barley root tissues increased when *P. indica* colonized the roots. Besides higher antioxidant enzyme levels, the ascorbic acid levels also increased [227], for which reason H_2_O_2_ and superoxide levels, as well as lipid peroxidation, were considerably reduced [228]. 

Drought also promotes malondialdehyde (MDA) accumulation [229]. Osmolytes are produced that maintain the sodium-potassium ratio [230]. Proline reduces tissue and cell damage [155,231,232], protects membranes and proteins from ROS damage and supports osmotic adjustment under abiotic stress [159,233]. Endophytes improve the drought resilience of plants by actively accumulating proline in the host tissue and lowering the MDA levels [234,235]. *P. tabulaeformis* seedlings treated with endophyte accumulated less MDA and enhanced proline accumulation [236]. Furthermore, under stress conditions, rice infected with *B. amyloliquefaciens* showed higher levels of the antioxidant amino acids cysteine, aspartic acid, glutamic acid, phenylalanine, and proline.

How endophytes stimulate the accumulation of antioxidants to counteract ROS in plants is not well understood. Microbial signals might directly control antioxidant enzyme levels at the transcriptional or posttranscriptional level to reduce ROS stress. Furthermore, ROS accumulation in the plant itself can be controlled by microbial signals. Takahashi et al. [237] showed that MKK3 and MAPK8 control ROS homeostasis in stress-exposed *Arabidopsis* plants. The MPK8 pathway negatively regulates ROS accumulation by controlling the expression of the *RBOHD* gene. The authors showed that Ca^2+^/CaMs and the MAP kinase phosphorylation cascade converge at MPK8 to monitor or maintain ROS homeostasis. One possible scenario could be that beneficial microbes counteract drought stress-induced ROS accumulation by activating the ROS-scavenging enzymes and the MPK8 phosphorylation module simultaneously, to directly restrict excess ROS formation. 

 *(h)* 
*Endophyte-regulated drought genes in the hosts*


Drought stress reprograms the gene expression profile [238]; the drought-responsive genes can be classified into two classes [180]. The first-class codes for transcription factor enzymes involved in phytohormone biosynthesis, as well as signaling compounds such as phosphatases and kinases. For example, *Paenibacillus polymyxa* increased transcript levels for the drought-response gene “*early response to dehydration 15*” in *A. thaliana*, which codes for a transcription factor that integrates different stress signaling pathways for improved drought tolerance [239]. The second class of genes is directly involved in protecting the cells from drought stress and includes genes for heat-shock proteins (HSPs), dehydrins, senescence-related proteins, membrane and cell wall protectants, late embryogenesis-abundant (LEA) proteins, osmoprotectants, transporters, and antioxidant enzymes [240,241,242]. 

Endophytic association also stimulates the expression of genes that maintain the cell wall structure intact under drought stress. The microbes promote the expression of host genes that manufacture cellulose, hemicellulose, and pectin, cell wall modifiers, and cell wall proteins for the host cell wall [217].

The endophyte *B. phytofirmans* PsJN enhances the expression of genes involved in cellular homeostasis under drought stress in potato plants [54]. Maintaining cellular homeostasis and balancing growth and defense/stress responses is regulated at multiple levels, with one level involving phytohormones. In plants colonized by *P. fluorescens*, the upregulation of auxin-response genes and the downregulation of ethylene-response genes is mediated by microbial control of the two plant phytohormone levels [243]. Plants colonized by the ACC deaminase-producing strain *Enterobacter cloacae* showed increased transcript levels for proteins involved in cell division and proliferation [244]. *P. chlororaphis*-colonized *A. thaliana* showed an increased expression of SA- and ethylene-responsive genes [245]. These examples demonstrate that hormones from both partners counteract drought stress to maintain cellular homeostasis in the host. 

Genes involved in the biosynthesis of secondary metabolites also play an important role in drought-tolerance responses. The *P. indica* colonization of soybean plants stimulated the phenylpropanoid and lignin pathway genes, both of which are known to play a role in oxidative stress tolerance [246]. Secondary metabolites participate in stabilizing the cell wall, function as stress-responsive components, or stabilize membranes. The *Pseudomonas putida* strain FBKV2-colonized maize seedlings demonstrated the upregulation of the genes involved in β-alanine and choline biosynthesis under drought stress [247]. 

 *(i)* 
*Epigenetic effects, small RNAs, and silent gene clusters in symbiotic interactions*


Arbuscular mycorrhizal fungi have strong effects on the epigenetics of their hosts, and vice versa. For instance, *Funneliformis mosseae* alters the DNA methylation profile in *Geranium robertianum* [248], while the DNA adenine methylation pattern was altered in *Mesorhizobium loti* during the colonization of different host plants [249]. Overall, DNA methylation events were observed in different endosymbiotic interactions [250,251]; Lidia et al. [252] concluded that endophytes can change the DNA methylation of wheat plants to enhance resistance against abiotic stresses. To what extent epigenetic alterations in plants are induced by colonizing microbes and which of them may contribute to drought tolerance is an open field for future investigations. Furthermore, this symbiosis can lead to the activation of new gene clusters in the microbiome (cf. [253]); it can be envisioned that they are involved in the biosynthesis of natural products involved in drought tolerance. The current investigations focus on the silent gene clusters of pathogens, alongside their activation and function after plant infection [254]. For instance, Motoyama [255] investigated the rice-blast fungus, *Pyricularia oryzae*, and identified secondary metabolites such as melanin, a polyketide compound required for rice infection, pyriculols, phytotoxic polyketide compounds, nectriapyrones, antibacterial polyketide compounds, produced mainly by symbiotic fungi including endophytes and plant pathogens, and tenuazonic acid, a well-known mycotoxin produced by various plant pathogenic fungi. Several reviews also discuss the beneficial microbes. Ribeiro et al. [256] investigated the metabolite profiles produced by Brazilian endophytic fungi, which do not cause harm to their hosts. Although the investigations did not identify stress- or drought-related compounds, the large number of new compounds may indicate that some of them might have drought-tolerance-related functions. Furthermore, Deshmukh et al. [257] describe antifungal compounds, including volatile organic compounds, isolated from fungal endophytes of medicinal plants, and some of them derive from the activation of silent biosynthetic genes during symbiosis. Dinesh et al. [258] describe secondary metabolites from silenced biosynthetic gene clusters of endophytic actinobacteria, and discuss the potential of these endophytes in the agro-environment as promising biological candidates for the inhibition of phytopathogens; the way forward is to thoroughly exploit this unique microbial community by inducing the expression of a cryptic gene cluster for encoding unseen products with novel therapeutic properties. These few examples from the literature suggest that beneficial endophytes might also produce relevant drought-tolerant compounds by the activation of silent genes under drought stress in symbiotic interactions. 

The same procedures hold true for the small RNAs that have been intensively studied in mycorrhizal interactions and numerous plant stress responses. These small non-coding RNAs play an important role in post-transcriptional gene regulation, during plant development, and in the responses to biotic and abiotic stresses and symbiotic interaction (e.g., [259,260,261]). They can be even transferred across species borders. It will be important to identify those miRNAs in plants that confer drought tolerance and respond to symbiotic microorganisms.

 *(j)* 
*Metabolic responses of endophytes in plants during drought stress*


Endophytes induce the production of novel metabolites, such as amino acids, non-structural carbohydrates, and various phenolic and alkaloid chemicals in the host, which accumulate to increase host fitness and water relations under stressful conditions [66,235]. Under water-deficit conditions, metabolome profiles in *Festuca arundinacea* (tall fescue) that was colonized with *E. coenophiala* revealed that the endophyte had a considerable impact on primary and secondary metabolism [235,262]; stress-exposed colonized plants accumulated additional sugars, sugar alcohols, amino acids, and mineral ions when compared to the uncolonized control [263]. Sugar accumulation is linked to the plant’s water-stress tolerance [264]. Various sugars can operate as regulatory molecules in the multiple signaling pathways [265] involved in maintaining redox balance and ROS scavengers and, thereby, participate in regulating water uptake and maintaining cell turgor under water-deficit environments [266,267].

Dastogeer et al. [268] showed that endophytic plants accumulated more osmotically active carbohydrates in their leaves than did non-endophytic plants. The presence of endophytes increased arabinose, mannose, and sucrose in *Nicotiana benthamiana*, as well as a few other sugars. These endophytes also improved water-stress tolerance in tall fescue, maize, and grapevine plants due to the higher and faster accumulation of stress-related metabolites, which is similar to reports for other plant–endophyte systems [218,235,269].

Redman et al. [115] reported that endophyte infection reduced the levels of particular amino acids in grasses, and the effects were dependent on host traits and nutrition supply. Nagabhyru et al. [235] found an altered amino acid profile in endophytic plants under water constraints that differed from the profile of the uncolonized controls. Increased levels of tricarboxylic acid-cycle intermediates, such as aconitate, citrate, fumarate, and malate represent another method for the plants to cope with water stress. Larger amounts of aconitate, fumarate, and succinate in endophytic plants indicate improved mitochondrial activity, resulting in increased amounts of reducing agents and ATP generation [270], although it is not clear how the levels of these molecules are regulated by endophytes. In comparison with non-inoculated plants, Gagné-Bourque et al. [271] found that *Bacillus subtilis* B26 sped up the timothy responses to drought stress by boosting the accumulation of either acquired or inducible metabolites that are linked with drought protection. Increases in soluble carbohydrates, including sucrose, fructans, and glucose were directly connected to the presence of B26 during drought conditions. Under water stress, the bacterized timothy produced considerable amounts of asparagine in the shoots, alongside valine, leucine, and isoleucine in the shoots and roots. Although many reports showed that endophytes stimulated, or rapidly activated, the accumulation of solute metabolites during stress, one has to keep in mind that the effects of fungal endophytes on plants are context-dependent [114,149]. Besides an overall stimulation of the osmotic potential by higher levels of soluble sugars in the cell, an understanding of the specific responses of a given symbiotic interaction might be helpful for potential agricultural applications. 

### 1.8. Systemic Distribution of Drought Stress Information in the Plant Body

Many endophytes colonize the roots either preferentially or exclusively. Since drought tolerance is often observed in the aerial parts of the plant, there must be an information flow from the roots to the leaves. Likewise, plants that are not colonized by endophytes must first recognize water-deficit conditions in roots; several molecular signals must then move from roots to shoots [272]. This requires “inter-organ signaling.” In the aerial parts, the information is further distributed between the distal tissues in each organ via “inter-tissue signaling” [70]. This raises the question of what compounds travel shootward. Are the plant-derived drought signals that travel to the leaves different from the signals that are activated by root-colonizing endophytes? The beneficial microbes can also support or accelerate an information transfer to the shoots, which already functions without endophytes. Since the vascular system of plants connects the roots and shoots, the signaling molecules most likely travel via these organ connections (if volatiles are not considered). Answers to these questions are important for agriculture since the harvested material is often aboveground. 

Hormones, in particular ABA, mRNAs, hydraulic signals, Ca^2+^, electric, and ROS waves, as well as hormones such as peptides, are candidates that can travel long distances to activate drought stress responses in the distal tissues and cells (summarized by Kuromori et al. [70]). Hormones and peptides likely travel through the vascular tissue. Takahashi et al. [272] and Kuromori et al. [70] summarized the signaling compounds involved in the systemic spread of drought information in plants. Here, we discuss two of these processes because they might be influenced by root-colonizing endophytes [11].

Upon the occurrence of drought conditions, ABA is synthesized in the roots and is translocated to the shoots via the xylem [273]. Although ABA is also synthesized in the aerial parts of the drought-exposed plants, systemic information flow requires (a) the synthesis of ABA or (b) the release of ABA from conjugates in the roots, then (c) its transport into the xylem and further to the guard cells, for instance, where it activates stomata closure and subsequent changes in the gene expression profiles. Synthesis occurs mainly in the vascular tissues, which are the major sites of ABA biosynthesis, while the contribution of ABA, which is released from its storage form, ABA glucose ester, is not completely understood. ABA is then exported from the cells via plasma membrane-localized transporters, and the *Arabidopsis* ABCG25, ABCG40, NPF4.6, and DTX50 are mainly functional in ABA distribution in the aerial parts (cf. the references in Kuromori et al. [70]). Thus, ABA is transported to its place of action and is then imported into cells, where it is sensed via the ABA receptors [70].

Many reports demonstrate that root-colonizing endophytes reduce the ABA levels in their hosts under drought or related stresses. Other reports demonstrate elevated levels, or changes, or tissue-specific ABA levels in colonized plants [274,275]. Since ABA is a component downstream of drought perception, lower ABA levels in colonized plants suggest that they suffer less under drought stress. Possibly, other compounds upstream of ABA are targeted by the endophytes, which reduces the drought stress and, thus, ABA production. Xu et al. [274] compared ABA levels in *P. indica*-colonized *Arabidopsis* plants during different developmental stages and interaction phases. In their analyses, the ABA level in the host has a strong influence from the symbiotic interaction, as also reported for other systems (cf. [275]). In the presence of a beneficial endophyte and an external stress such as drought, the plants must decide whether they invest in either stress-tolerance responses or symbiotic features. Stec et al. [276] proposed that the fluctuation in ABA levels may work as an alert system that calculates the ratio between costs and incomes. The ABA level may function as a monitor for the decision as to whether an investment in a particular symbiotic interaction is greater than the profit it would bring. In light of such a hypothesis, the ABA level in the plant is established by the plant in response to the drought stress. An endophyte represses its concentration in the host when the symbiotic plant is better protected against drought than the host alone. The propagation of the ABA information is coupled with signals from the microbes. It might be an important task in the future to understand whether root-colonizing endophytes influence local and systemic ABA levels only in the roots, or whether microbe-derived signals also interfere with ABA levels after traveling to the shoots. Vahabi et al. [85] demonstrated that ABA can also be used by *P. indica* for the long-distance propagation of biotic stress information. Numerous other hormones are also involved in drought stress responses, and the scenario might be similar for them also [70]. Among these hormones is JA, which is important for induced systemic resistance (ISR). ISR enhances the defense systems of the plant and is effective against necrotrophic microorganisms and insect herbivores, which often attack the aerial parts of the plants. The ISR operates systemically and is activated by plant growth-promoting bacteria or fungi that are often associated with the roots. Only a few studies have been performed so far that link these biotic resistance mechanisms to abiotic stress tolerance (e.g., [277,278,279,280,281] and the references therein). 

Furthermore, small peptides such as CLE25 and CLE9 are ideal candidates for long-distance signaling (cf. above). Takahashi et al. [87] report that the peptide CLE25, together with the BAM1 and BAM3 LRR receptor-like kinases are involved in root-to-shoot communication during dehydration stress in *Arabidopsis*. The *CLE25* gene is expressed in vascular tissues and is enhanced in roots, in response to dehydration stress. The root-derived CLE25 peptide moves from the roots to the leaves, where it induces stomatal closure by modulating ABA accumulation and thereby enhances resistance to dehydration stress. BAM receptors are required for the CLE25 peptide-induced dehydration stress response in leaves; the CLE25-BAM module, therefore, probably functions as one of the signaling molecules for long-distance signaling in the dehydration response [87]. CLE25 could be transported in the vascular tissue; however, the way in which this occurs is not fully understood. Furthermore, the multiple functions of the CLE peptides make it difficult to predict whether or how they are influenced by endophytes. CLE25 might only be the tip of the iceberg. Other peptides, such as CLE9, CLE10, phytosulfokine precursors, and the subtilisin-like protease also respond to drought stress (cf.[70]) and are potential targets for endophytic signals. Moreover, the genome of *Arabidopsis* contains more than 7000 small open reading frames with largely unknown functions. 

### 1.9. Crop Growth Modulation Using Fungal Endophytes

The balanced interactions of endophytes with their host plants during the entire symbiotic phase allow for improved adaptation to environmental changes [282]. However, the effectiveness of these microbes at the field level is highly dependent on their performance, which necessitates additional research into the barriers to effective product development.

Interestingly, there is a vast amount of research on beneficial microbes for plants that may be utilized to guide the screening process, develop best practices for validation, and uncover some problems that may prevent these benefits from being transferred from greenhouse to field. Numerous publications support the beneficial effects of fungal endophytes on plant development and performance in adverse conditions [283]. In the United States, Australia, and New Zealand, *Epichloe* endophyte strains have been employed to boost the productivity of forage grasses in the field and the robustness of turf grasses [284,285]. Endophyte-mediated plant trait improvement provided roughly NZD 200 million per year to the New Zealand economy [286]. *Trichoderma* sp., which lives on the roots of stressed plants, improved yields in field studies [287]. *Fusarium equiseti* increased *T. subterraneum* herbage yield in the greenhouse, while *B. spectabilis* improved *T. subterraneum* forage quality by reducing the fiber content and *P. pratensis* fodder quality by increasing the crude protein content. *S. intermedia* increased Ca, Cu, Mn, Pb, Tl, and Zn mineral uptake in sub-clover, while *M. hiemalis* increased K and Sr uptake in Kentucky bluegrass, demonstrating the potential of fungal endophytes to improve herbage productivity and the nutritional value of fodder [288]. Despite the immense potential of endophytes in abiotic stress tolerance responses, field-rational tests are few. 

## 2. Open Questions

Under drought stress, the biodiversity in soil decreased significantly but increased in the rhizosphere community [289]. The ratio of fungi to bacteria is an important index for measuring the microbial community structure. All plants are equipped with a core microbiome, which raises the question of what additional microbes can colonize the plants to confer drought resistance. Under drought stress, the root-associated fungal community changes, and this, again, is dependent on the genotype, as shown for rice (cf. [289], and the references therein). Fungi that help to improve the drought tolerance of rice also change their secretome patterns under drought. The rhizobacteria can produce plant hormones or inhibit plant hormone production, which may directly affect plant growth, while fungi might play an important role in nutrient acquisition under drought (cf. [289]). The microbial community in the drought-exposed host is probably crucial for the stress resistance of the host. 

How do the microbes enter the plant and how do they compete with other microbes in the host tissue? Very little is known about the acceptance of new members in the plant microbiome and their interaction. They might be differences between the entry mechanisms for bacteria and fungi. *Rhizobia* are widely studied and their entrance via root hairs is probably the best-understood mechanism. Alternatively, entry via intercellular invasion has been observed in many legumes. Although there are common features that are shared by intercellular infection mechanisms, differences are observed in the site of root invasion and bacterial spread on the cortex, reaching and infecting a susceptible cell to form a nodule [290]. Considering root endophytic fungi, Dong et al. [291] have shown that the maturation zone is the main target for *P. indica* for entering Chinese cabbage roots. Tseng et al. [292] showed that the hyphae of a *Trichoderma* strain invaded the root hair of *Arabidopsis*, and conidiophores were found at the tip of the root hair. The vast majority of the endophytic bacteria and fungi live in the apoplastic space of their hosts. If the microbe alters the functions within the host cells, chemical mediators from the microbe must cross the host plasma membrane, or microbial signals are perceived in the apoplastic space and transduced into the host cytoplasm, or transport processes across the host plasma membrane are altered. More research is required to understand the transport processes across membranes. 

Finally, many agriculturally interesting endophytes come from desert plants. The response of a desert plant to drought stress differs fundamentally from plants that suffer under drought, but they can otherwise perform all cellular, biochemical and developmental processes, albeit at reduced rates, under the extreme conditions in the desert (cf. above). The complete absence of water over longer periods of time in the desert, often combined with permanent nutrient limitations due to the pure soil/sand conditions, results in an entirely new strategy for the survival of the plant. The supporting role of their associated endophytes is probably also different from the role that endophytes play for plants suffering under water limitations. Therefore, laboratory studies are necessary before large-scale agricultural applications are planned. These may uncover completely new mechanisms of how plants in symbiotic interaction with endophytes deal with water and/or nutrient shortage.

Finally, the introduction of new endophytes into a new soil may influence the soil microbiome, which needs to be considered for large-scale applications.

## 3. Conclusions

Exploiting endophytes to increase agricultural productivity is a fascinating prospect. A successful symbiosis is only stable if both partners profit from the interaction. Therefore, the search for candidates for agriculturally relevant endophytes should be straightforward in terms of field experiments. Second, endophyte-induced drought tolerance in the crop plant is the result of multiple processes that are activated in the host. The activated processes and their combinations are specific for each microbe and plant/microbe interaction. The task for scientists is to understand the molecular and biochemical basis of these mechanisms and their crosstalk, whereas the task of agriculture is the identification of microbes that best fit the given environmental conditions. The newly introduced microbes must compete or interact with the local soil microbiome; therefore, understanding how beneficial endophytes are recruited and maintained within the host at different growth phases is crucial. Weather is difficult to predict and requires the adaption of the plants to a wide range of threats. The broad spectrum of endophyte-induced responses in the hosts provides better opportunities for successful adaptation than the application of chemicals or fertilizer or the use of transgenic material that mainly targets one or a low number of pathways in the plants. Endophytic consortia from extremely drought-stressed regions, which are already stress-tolerant by themselves, may have the potential to transfer these features to new hosts/crops that are not drought tolerant per se. Besides not always being successful, our understanding of the mechanism of drought tolerance in the new host might be very helpful because the endophyte may use quite different and unique methods. The introduction of endophytes into a new ecosystem can be beneficial or non-beneficial for soil health. The incorporation of microbes is an attractive and low-cost method for sustainable agriculture and has several advantages over genetically modified plants or the use of fertilizers, as a symbiotic interaction can respond more flexibly to environmental changes.

## Figures and Tables

**Figure 1 ijms-23-09194-f001:**
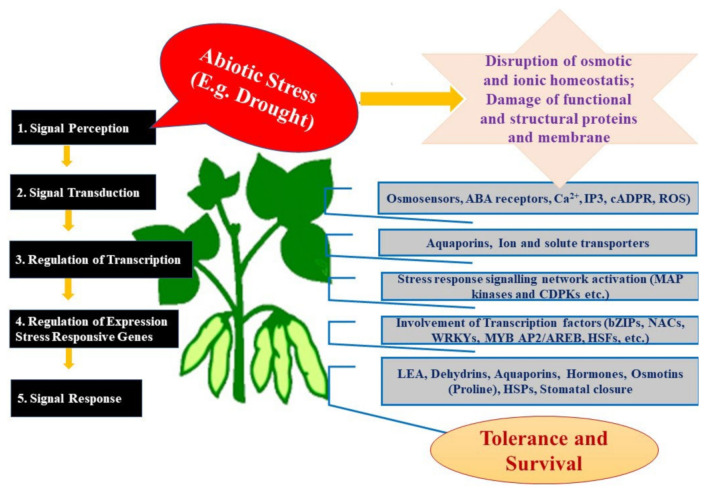
The response of plants to drought stress. The plant’s internal structure under drought stress is schematically described, showing the intracellular signal transduction pathways, along with the molecular regulation mechanism of plants. Drought-regulating substances, reactive oxygen species (ROS), and scavenging processes in plants are shown. An analysis of physiological and biochemical responses relevant to osmotic regulation metabolism, drought-induced protein metabolism, and reactive oxygen metabolism is shown (modified from Lamaoui et al. [102]; Zenda et al. [103]; Yang et al. [78]).

**Figure 2 ijms-23-09194-f002:**
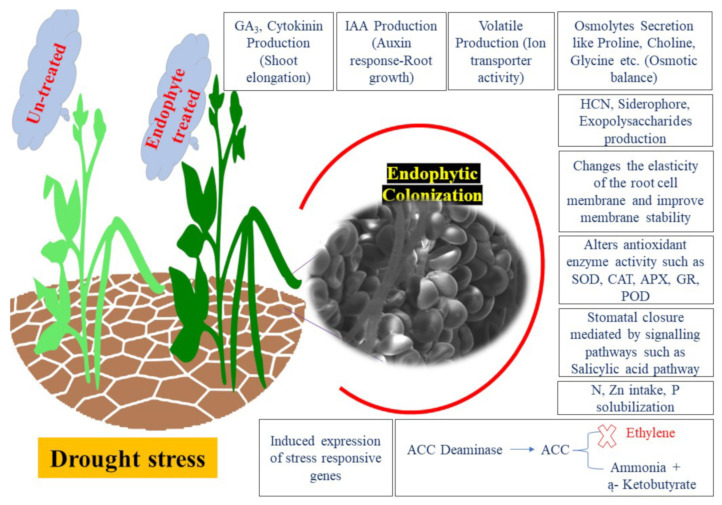
Endophyte-mediated drought tolerance in plants. As part of a plant–endophyte association, the latter employs a variety of strategies to mitigate the abiotic stresses from the host’s natural habitat. A lack of nutrients in the habitat can be countered by mechanisms such as the production of growth-promoting hormones, other compounds, and nitrogen fixation by plant endophytes. As an additional benefit, some endophytes can go a step further and provide host plants with crucial protection against harsh environmental conditions by activating antioxidant enzymes and the production of stress-responsive molecules (modified from Vurukonda et al. [158]; Ullah et al. [159]; Verma et al. [160]).

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
