# Peer review of "Is Endophytic Colonization of Host Plants a Method of Alleviating Drought Stress? Conceptualizing the Hidden World of Endophytes"

_ijms, 2022, doi:10.3390/ijms23169194_

Round 1

Reviewer 1 Report

My review of the manuscript of “Is endophytic colonization of host plants a method of alleviating drought stress? Conceptualizing the hidden world of endophytes (Manuscript ID: ijms-1848258) is as follows:

This review article presents the importance of endophytic microorganisms in the drought tolerance of crops. This review article presents the importance of endophytic microorganisms in the drought tolerance of crop plants, from plant development to fruit ripening. Agriculture is extremely exposed to weather extremites. As a result of climate change, the average temperature of our Earth increases and the frequency of extreme weather conditions, including those of drought years. Accordingly, the topic of this review article is extremely timely. The number of citations is enormous (304), the proportion of newer articles (2018-2022) on the topic are sufficient, 34%. However, this review article is unfortunately rather poorly written. The lines are not numbered, which makes it difficult to review (especially for such a large article), so it is difficult to draw the authors' attention to minor errors. The figures and tables must be inserted in the appropriate place (after the first citationss in the main text cf.: instructions for authors/preparing figs, schemes and tables). Unfortunately, this reveals that the authors were inattentive in formally editing the manuscript, even though the Journal provides its own template format for this. Therefore, I am only forced to highlight the major things. The introduction is well organized, e. g. I like that the authors highlight the issue “problematic” of gene modification (page 3.). But the topic and purpose of the article should be better formulated and described in more detail at the end of the introduction. At the moment there is only one sentence about it. A bigger problem than these formal matters is that the manuscript is not clear, it grabs too much and takes too little. There is a lot of redundant and unnecessary information in it (e.g. I don't think it's necessary to explain the SPAD measurement, it's irrelevant from the point of view of the article, but there are a few more such examples.). We would expect this review article to provide clear problem statements and their (possible) solutions, illustrated through examples and a more focused approach would be better. I am missing a table(s) showing the endophytic species, their effect and on which plant species was tested.

For the reasons mentioned above, I unfortunately have to reject the current form of the manuscript.

Reviewer 2 Report

Dear Authors,

You did a very good job, but as a reviewer, I am obliged to point out my doubts and reservations.

In many fragments of the text, you indicate the unconditional relationship between the endophyte and the host, suggesting that some relations must take place in the presence of an endophyte. As you know, it is not so, the endophyte-host relationship is very variable, which you mention only in passing. It is therefore worth revising the text of your publication in this regard.

Below are detailed comments on the indicated fragments of the text:

Page 4, lines 4 – 5 from the top: It is not always the truth. Endophytes applied to seeds MAY enhance etc..

Page 4, line 8 from the top: Like above. Under drought conditions, the endophytes MAY boost ...etc.,

Page 7: Table 1: And what about grasses (Poaceae) ? There are many publications dealing with drought assistance to grasses by fungal endophytes. Please, add some...

 Page 9 and 10: the whole page 9 and 3 lines from the top of page 10 : Quite interesting but only at the very end some relations to endophyte interaction, Try to condense/re-write ….

 Page 11 (last paragraph) and Page 12 (first 7 lines from the top) - Considering the last sentence ("Whether of how ....(..)), it is quite discussible to put all before. If this process is unknown, why mention it?

 Page 12, 8 – 11 lines from the top: But nothing about relations with endophytes....

 Page 14: Better try to cite publications, not books (ref. no.109)

 Page 15: 15 text line from the top: “Numerous studies..” Which studies, where are citations?

 Page 20: 6 text line from the top: ‘endophytes’ not ‘microbes’.

 Page 22: 2 text line from the bottom and page 23 second paragraph from the top: ‘P’ not ‘Pi’

 Page 30: 3 line from the top: ‘B26 speed’ not ‘B26 sped’

 Page 41: ref. 104: Labudda, M., Lipid peroxidation as a biochemical marker for oxidative stress during drought. An effective tool for plant breeding. 2013.” – is this an article or book?

Round 2

Reviewer 1 Report

Compared to the original version, the structure, language and content of the manuscript have been significantly improved. I would like to draw the attention of the authors to a few shortcomings.

Line 4: The numbering of the authors' affiliation should be indexed.

Line 10: remove "In" bold font ("In the wake of changing climatic...")

Line 5-121. The font size is not correct.

Line 119-121. If available, the table can be supplemented with several references. (Table 1.)

Line 911-1634. Please check the format of the References (font) and whether they they are cited in the text. Remove unnecessary references.

Reviewer 2 Report

Dear Authors,

Thank you very much for taking my remarks into consideration.

Now, in my opinion, your text is much more comunicative.

Sincerely

Reviewer

Author Response

Please seed the attachment
